# Components and Architecture of the Rhabdovirus Ribonucleoprotein Complex

**DOI:** 10.3390/v12090959

**Published:** 2020-08-29

**Authors:** Christiane Riedel, Alexandru A. Hennrich, Karl-Klaus Conzelmann

**Affiliations:** 1Institute of Virology, Department of Pathobiology, University of Veterinary Medicine Vienna, 1210 Vienna, Austria; 2Max von Pettenkofer-Institute Virology, Faculty of Medicine, and Gene Center, LMU Munich, 81377 Munich, Germany; hennrich@genzentrum.lmu.de (A.A.H.); conzelmann@genzentrum.lmu.de (K.-K.C.)

**Keywords:** rhabdoviruses, ribonucleoprotein structure, rabies virus, vesicular stomatitis virus, lagos bat virus, lyssavirus, vesiculovirus

## Abstract

Rhabdoviruses, as single-stranded, negative-sense RNA viruses within the order *Mononegavirales*, are characterised by bullet-shaped or bacteroid particles that contain a helical ribonucleoprotein complex (RNP). Here, we review the components of the RNP and its higher-order structural assembly.

## 1. Introduction

Rhabdoviruses (family *Rhabdoviridae*, order *Mononegavirales*) are negative-sense, single-stranded RNA viruses with a genome of about 10.8–16.1 kb. The prototypic mammalian rhabdoviruses include the rabies virus (RABV) and vesicular stomatitis virus (VSV), which cause severe human and animal disease. The cell biology of these viruses is highly divergent in many important aspects. VSV is transmitted in nature by insect vectors, causes a rapid general host cell shutdown in mammalian cells and replicates to very high titres. The neurotropic rabies virus is directly transmitted between mammals, grows slowly and relies on stealth strategies to escape host responses. Despite having divergent biology, the viruses share a highly similar genetic and structural organisation.

The small genome encodes the minimal set of five proteins that are all present in the virus particle. They consist of the nucleoprotein (N), the matrix protein (M), the surface glycoprotein (G), the phosphoprotein or polymerase cofactor (P) and the RNA-dependent RNA polymerase (L). In contrast to other members of the mononegavirales, such as paramyxoviruses or filoviruses, rhabdoviruses’ enveloped particles are characterised by a bullet-like or bacteroid appearance (Figure 1). The G-protein is the only surface glycoprotein, and therefore the essential factor for virus attachment and entry, including fusion. It belongs to the class III of fusion proteins, together with herpesvirus gB or baculovirus GP64 protein (as reviewed in [1]). Inside the viral envelope sits a helical ribonucleoprotein complex (RNP), which ends with a cone that has tighter helical turns (Figure 1). The RNP’s main building blocks are the N- and M-proteins. Both are structurally highly conserved between the genera *Vesiculovirus* (exemplified by VSV) and *Lyssavirus* (including RABV and Lagos bat virus (LBV)) (RMSD (root mean square deviation) C_α_ 2.5 Å for N [2], 3.1 Å for M [3]), but the sequence conservation at the amino acid level is only 15% (N-protein). The structural homology implies the need for a specific protein layout to fulfil tasks linked to RNA binding and RNP generation. In contrast to N- and M-proteins, which are incorporated into the RNP at approximately equimolar amounts (≈1200:1200 based on a CryoEM reconstruction [4] and 1258:1826 based on biochemical analysis [5]), the P-protein is estimated at 466 copies per virion [5], while the estimated number of G-protein is 1205 copies per virion. With a molecular weight of 242 kD, and as the most important protein for genome replication and transcription, the L-protein is the least numerous protein present in the virion, with an estimated 50 copies per virion [5].

## 2. N Protein Structure and Function

The atomic structures of the N-proteins of VSV [8] and RABV [9] were resolved using X-ray crystallography (Figure 2). Both proteins were crystallised as rings in a complex with RNA, containing either 10 (VSV) or 11 (RABV) N-protein monomers. The monomer itself can be separated into an N- and C-terminal domain, which are separated by a cleft that is responsible for the binding of the RNA genome. Each N-protein is associated with nine nucleotides, and the N-RNA interaction is mainly coordinated by basic amino acid side chains that interact with the RNA’s phosphate groups. The corresponding five residues for RABV are Arg149, Arg168, Arg225, Arg323 and Arg434, while the corresponding six residues for VSV are Arg143, Arg146, Lys155, Lys286, Arg317 and Arg408. Only two of these basic residues - RABV Arg149/VSV Arg143 and RABV Arg434/VSV Arg408 - are structurally conserved between RABV and VSV, resulting in a different arrangement of the nucleotides in the RNA binding cavity [2]. Higher-order N-protein assemblies are stabilised by an N- and a C-terminally protruding extension, which interact with the neighbouring N-protein monomers to form a circular assembly. A tabular presentation of the interacting residues is given in [2].

N-protein is not only essential for RNP formation but is also an essential factor during replication and transcription, as the viral polymerase only processes RNA that is bound to N-protein. Currently, it is not known how the polymerase complex of L and P gains access to the RNA in complex with N-protein and how this process is coordinated.

## 3. M-Protein Structure and Function

As observed for the N-protein, the sequence conservation of the 20–25 kDa M-protein between the Lagos bat virus (LBV, genus Lyssavirus) and VSV is very low (<10%), whilst the structural conservation is substantial (RMSD of 139 equivalent C_α_ is 3.1 Å) [3] (Figure 3). The majority of the M-protein consists of a compact, globular structure dominated by a four-stranded β-sheet on one side and two α helices and a small, two-stranded β-sheet on the other [3,10,11]. The N-terminus of the protein (aa) 1–47 in LBV, aa 1–57 in VSV) is mostly unresolved in the crystal structure, except residues 30–37 (LBV) and 41–52 (VSV), which are buried in a hydrophobic pocket of the globular domain of the neighbouring protein [3]. This interaction leads to the formation of non-covalently linked, linear polymers [3].

The M-protein is a multifunctional, central player in the formation of rhabdovirus virions. It is responsible for the condensation of the nucleocapsid [12], which produces the final shape found in the virion [12,13]. The M-protein is indispensable for viruses to form and bud [14], and it interacts with the lipid membrane, G-protein and N-protein. The interaction with the membrane is mediated by electrostatic interactions [11,15,16]. The forces required for the budding process are hypothesised to be generated by M- and G-proteins in a so-called “push and pull” model [17,18]. A special role in budding has been assigned to the “late domain” motif (PPPY aa 24–27 VSV, PPEY aa 35–38 LBV) at the M-protein N-terminus. This motif has been shown to interact with Nedd4 and its ubiquitination is important for efficient budding [19,20,21,22,23,24]. Furthermore, an interaction of the first 10 N-terminal amino acids with dynamin is essential for virus assembly [25] and the N-terminus of the M-protein is also responsible for the interaction with the N-protein [26] and lipid membranes [27,28]. Apart from its roles in virus assembly, the VSV M-protein suppresses host mRNA transcription and translation by interfering with the export of mRNAs from the nucleus [10,29,30,31] and the phosphorylation of transcription factors [32,33,34,35]. Additionally, the VSV M-protein has been shown to affect immunoproteasome formation via interaction with LMP2 [36].

## 4. P-Protein Structure and Function

The P-protein (34 kD) is a central (co-)factor in the replication cycle of rhabdoviruses. It can be divided into four domains. These are the N-terminal domain (P_NTD,_ aa 1–90, RABV), which is disordered but adopts defined conformations upon interaction with N or L [37,38,39,40,41,42]; the central domain (P_CED,_ aa 91–130, RABV); the C-terminal intrinsically disordered domain (P_CID,_ aa 131–195, RABV); and the C-terminal domain (P_CD,_ aa 195–296, RABV). The atomic structures of P_CED_ (pdb: 2fqm [43], VSV; pdb: 3l32 [44], RABV), which mediates the dimerisation of the P-protein, and P_CD_ (pdb: 2k47 [45], VSV; pdb: 1vyi [46], RABV), which is important for interacting with the N-RNA complex and parts of the VSV P_NTD_ with N [37], have been resolved. P-protein forms a parallel dimer with an elongated shape, in which P_CED_ acts as a central core, from which four flexible structural elements emerge [47,48]. Within the cell and the virus, P is present in different N-terminally truncated forms [49,50,51], which are generated by alternative initiation sites, and different phosphorylation states [52,53]. Essential functions of the P-protein in the viral life cycle are numerous. The P-protein prevents the self-assembly and RNA-binding of the N-protein (monomeric N-protein not bound to RNA: N^0^), with one N-protein bound to one or two P-proteins via residues in P_NTD_ [54,55,56,57,58,59], thereby ensuring that the N-protein specifically binds to viral RNA but not viral mRNAs or cellular RNAs. The P-protein is also the essential but catalytically inactive cofactor of L-protein. P_NTD_ is the domain involved in the interaction with the L-protein [40,41,60,61]. This interaction is not essential for the enzymatic function of the L-protein, but rather for the generation of longer RNA molecules [62]. Binding to the N-RNA complex is mediated by P_CD_ [63,64]. The RABV P-protein is also an antagonist of the interferon system. The P-protein of RABV interferes with the phosphorylation of IRF3 [65], retains activated STAT1 and STAT2 in the cytoplasm [50,66] and interacts with TRIM19 [67,68,69]. Interactions of the RABV P-protein with additional cellular proteins have been described [39,70,71,72,73] but are beyond the scope of this review.

## 5. L-Protein Structure and Function

The L-protein is the enzymatically active component of the transcriptase and replicase complex (reviewed in [74]), with a molecular weight of 242 kD. The enzymatic activities of the L-protein include an RNA-dependent RNA-polymerase (RdRp, aa 35–865, positions are given for VSV) [6], a GDP polyribonucleotidyl transferase (aa 866–1334) [75], a methyltransferase (aa 1598–1892) [76,77,78,79] and it also acts as a polyadenylase [80]. It can be divided into three catalytic and two structural domains. The RdRp and the capping domain form the core of the structure. In the absence of P, the remaining three domains—connector domain, methyltransferase domain and C-terminal domain—have no fixed position regarding the core of the structure [6,40,41,81]. The P-protein segments that bind to the L-protein that can be structurally resolved are aa 49–56 (interaction with the C-terminal domain), 82–89 (binding between the C-terminal domain and RdRp) and 94–105 (binding between the connector and C-terminal domains) for VSV [40]. For RABV, a 37 aa segment of P-protein, ending with Glu87, could be identified as being bound to the connector, the RdRp and the C-terminal domain of L-protein [41].

## 6. Assembly of the RNP

During replication, newly generated virus genomes are coated with N-protein derived from N^0^–P complexes, with N^0^ referring to the monomeric N-protein. It has been shown that P_NTD_ interacts with N^0^ via hydrophobic residues and that the interaction interface is not located in the RNA binding groove [37]. The nucleocapsid is formed during replication. Subsequently, unbound M-, P- and L-proteins associate with the newly formed nucleocapsid to generate an RNP that is not yet condensed, as observed for the RNP inside virions. RNPs are transferred to the plasma membrane, where they are recruited to membrane microdomains containing M- and G-proteins at the budding sites [82,83,84,85]. Here, the RNP becomes condensed into a tight helix and is stabilised due to the activity of the M-protein. In vitro, it has been demonstrated that the exposure of an N-RNA complex to low pH and low ionic strength is sufficient for the formation of an RNP shape, as observed in the virion [86]. The 5′ end of the genome is located in the cone of the RNP [4]. The “pushing” action of the M-protein, which is the major budding factor, and the “pulling” action of the G-protein result in the budding of virus particles from the membrane [14,17,87]. VSV RNPs containing the genomic, negative-sense strand are preferentially packaged due to the trailer sequence at the 5′ end [88,89], while RABV packages both the + and − RNAs indiscriminately. The presence of 98% of negative-sense genomes in RABV particles is a result of the highly biased replication that favours the generation of negative-sense, genomic RNAs. To efficiently bud from the cell, rhabdoviruses rely on proteins of the ESCRT complex, which interact with the late-domain present in the N-terminus of the M-protein [22,90].

## 7. Morphology of the RNP in Virions

In the virion, the RNP is condensed into a left-handed helix [4,91] consisting of an inner helical ring of the N-RNA complex and an outer helical ring consisting of M-proteins. The N-terminal domain of the N-protein interacts with the M-protein, whilst the C-terminal domain is located towards the centre of the virion. The RNA-binding cavity of the N-protein, containing the viral genome, is oriented towards the RNP tip. Interactions between the N- and C-terminally protruding extensions of the N-protein connect neighbouring N-proteins along the helical turn. Due to the changes in the angle towards each other and the increased distance between the N-proteins in the CryoEM maps when compared to the arrangement in the crystal, additional stabilising interactions, as observed in the atomic structures of the N-protein of RABV and VSV between the N-protein C-terminal domains, should not be relevant for neighbouring N-proteins on the same helical turn in the RNP trunk [4,91]. However, for VSV, it has been proposed that these interactions are important contributors for the formation of the RNP tip, and it has been shown that mutagenesis of the involved residues (Ile237, Arg309, Tyr324 and Glu419) results in the formation of N-rings with an increased diameter [4]. Even though the structural similarity between RABV and VSV M- and N-proteins is striking, the architecture of the RNP of these two members of the family *Rhabdoviridae* differs substantially (Figure 4). Connections between the helical turns are established between N- and M-proteins in the RNP reconstruction of RABV, and between N- and M-proteins in the RNP reconstruction of VSV. Therefore, the M-protein forms a 3D lattice on the outside of the N-protein-RNA helix in VSV, whereas in RABV, only interactions between neighbouring M-proteins on the same helical turn can be observed. For the VSV M-protein, the CryoEM model suggests that the M-protein N-terminus, which is not resolved in the crystal structure, is responsible for all interactions with neighbouring M- and N-proteins. The resolution of the RABV RNP reconstruction does not allow for unambiguously assigning a certain position of the crystal structure in the CryoEM density map, but still implies that the N-terminus of the M-protein (or parts thereof) mediates the interaction with the N-protein on the neighbouring helical turn. An additional architectural difference is the N-protein helix orientation relative to the central axis of the virion. This angle is 63° in VSV and 44° in RABV. Furthermore, the helical turns in the RABV RNP are 20 Å further apart than in VSV. For VSV, the number of N-proteins per helical turn in the trunk of the RNP was determined to be 37.5. N-proteins are assembled in such a manner that the N-proteins of one helical turn are exactly positioned between N-proteins of the following turn, causing the helix to repeat itself every second turn. This repeating pattern could not be demonstrated in the reconstruction of the RABV RNP, where the arrangement of neighbouring helical turns suggests a repeat every 4–6 turns. 

## 8. Conclusions

Rhabdoviruses possess a characteristic RNP of considerable size, given the small RNA genome. Its architecture, consisting of building blocks that are highly conserved on the structural level but quite divergent at the amino acid level is comparable between VSV and RABV. However, despite the highly homologous building block structure, significant differences can also be observed, especially regarding the helical arrangement, helical repeat and interactions between helical turns. Therefore, additional research efforts are warranted to generate higher-resolution RNP structures to be able to clearly assign and functionally define the M-protein N-terminus, RNA-N-protein interactions inside the virion and the location and amount of L- and P-proteins as minor components of the RNP.

## Figures and Tables

**Figure 1 viruses-12-00959-f001:**
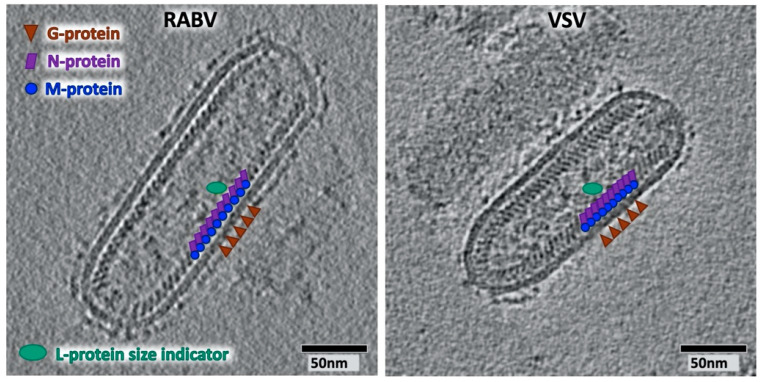
Cryo-electron tomograms of rabies virus (RABV) and vesicular stomatitis virus (VSV) particles at the same scale. The locations of the M- and N-proteins are indicated by blue circles and purple rectangles, respectively. The location of the G-protein is indicated by red triangles, and the size of the L-protein, as derived from the electron density map of the VSV L-protein generated by [6], is indicated in green. The tomograms were acquired on an FEI Glacios (ThermoFisher, OR, USA) with a Falcon2 (ThermoFisher, OR, USA) direct electron detector, reconstructed in etomo [7] and visualised by employing 3dmod [7].

**Figure 2 viruses-12-00959-f002:**
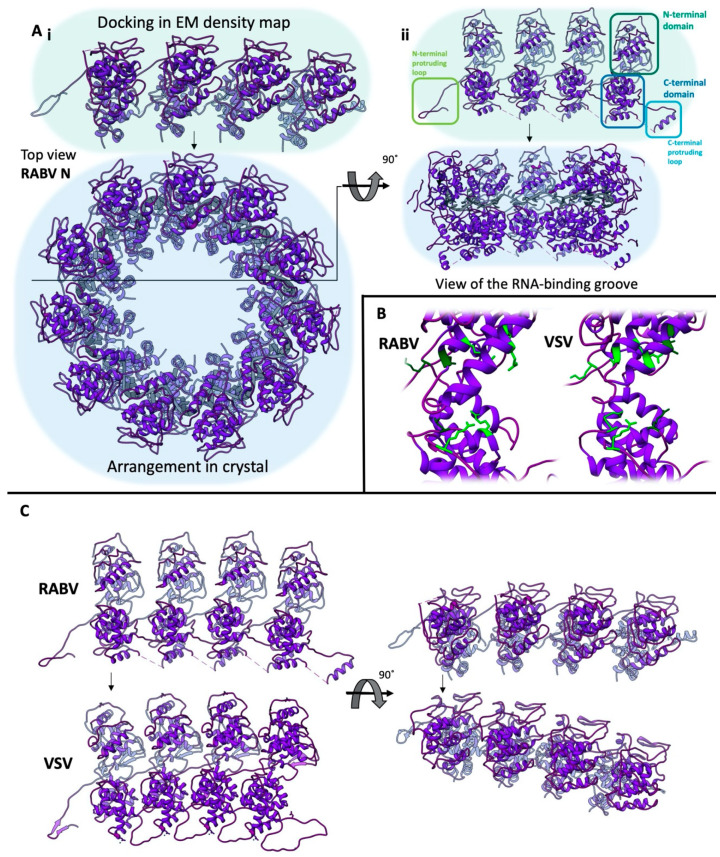
Comparison of the RABV and VSV N-protein structures (pdb 2gtt (RABV, [9]) and 2gic (VSV, [8])). (**A**) Comparison between the arrangement of the RABV N-protein in the crystal and docked in the cryo-electron density map from the top (**i**, view on the N-terminal domain) or the RNA binding groove (**ii**). Panels i and ii are not at the same scale. (**B**) Comparison of the residues interacting with the phosphate backbone (light green) of the viral genome between RABV and VSV. Positively charged residues that do not interact with the phosphate backbone but protrude into the RNA binding groove are depicted in dark green. (**C**) Comparison of the arrangement of N-proteins when docked into the CryoEM reconstructions of the RABV and VSV ribonucleoprotein complex (RNP). The N-monomers used for alignment of the different N-protein arrangements are connected using black arrows. EM: electron microscopy.

**Figure 3 viruses-12-00959-f003:**
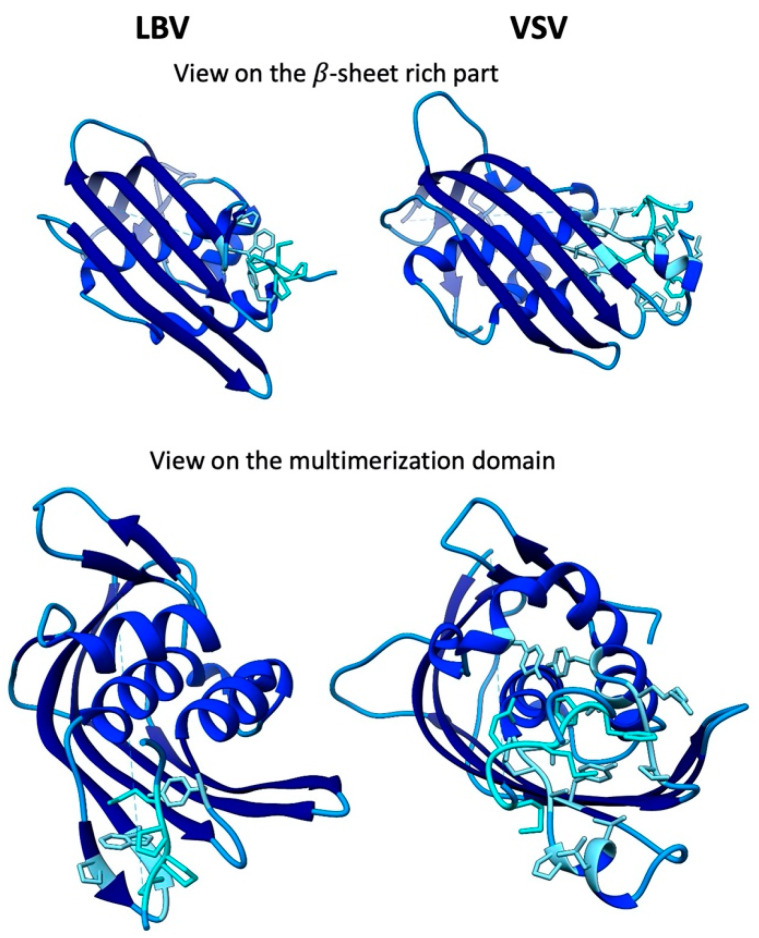
Comparison of the structure of the Lagos bat virus (LBV) (pdb 2w2s) and VSV (pdb 2w2r) M-proteins [3]. Residues involved in the interaction between the M-proteins are depicted in cyan on the N-terminal fragment and in light blue on the C-terminus.

**Figure 4 viruses-12-00959-f004:**
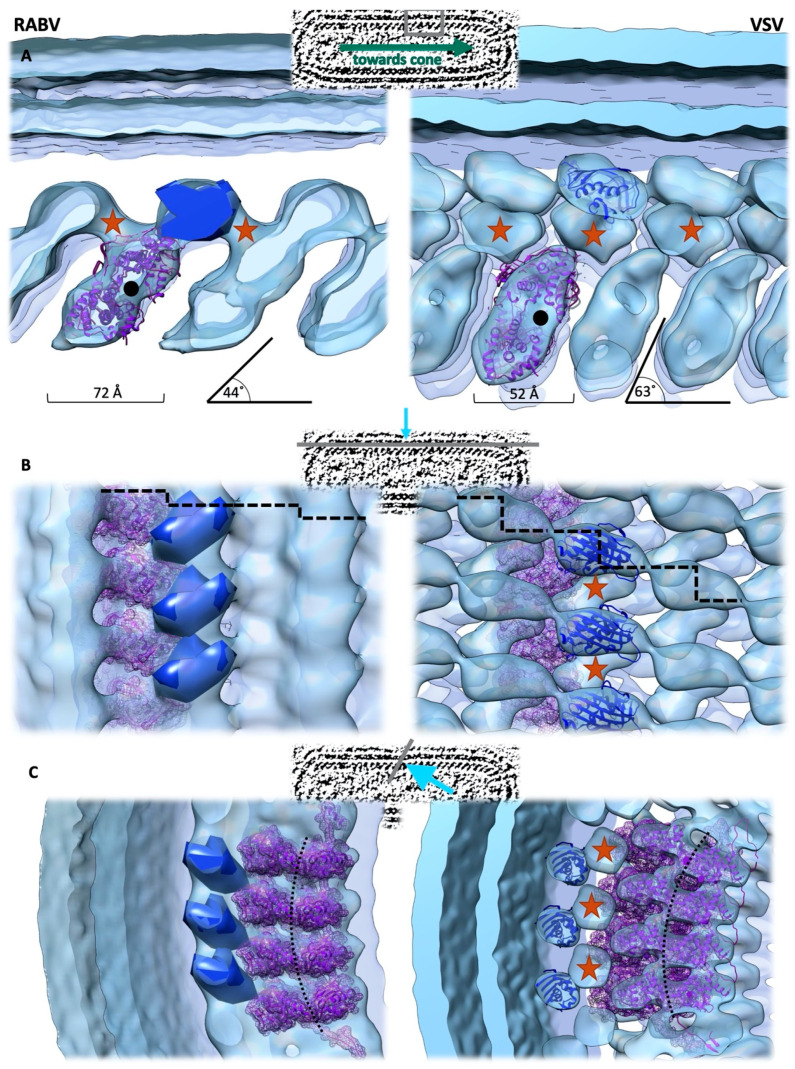
Comparison of the electron density maps of the RNP of RABV (EMD-4995) and VSV (EMD-1663). The viewing direction is indicated above each panel, where: (**A**) a view from the side, (**B**) a view from the membrane and (**C**) a view towards the RNA binding groove. The distance between the helical turns, as well as the deviation of the helical turns from the central axis of the virion, are indicated. The location of the genome is indicated by a black dot or a black dotted line. The N-protein structures (pdb 2gtt (RABV), 2gic (VSV)) are docked in the corresponding densities and shown in purple. The VSV M-protein structures (2w2r, blue) are docked in the orientation, as proposed by [4], whilst the location, but not the orientation, of the RABV M-protein C-terminal domain is depicted as a blue density. The proposed location of the M-protein N-terminus is indicated by red stars. Dashed lines in the middle panel indicate the step to the corresponding M-protein on the next helical turn.

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
