# Peer review of "Components and Architecture of the Rhabdovirus Ribonucleoprotein Complex"

_viruses, 2020, doi:10.3390/v12090959_

Round 1

Reviewer 1 Report

This manuscript is a brief review discussing differences between the ribonucleoprotein (RNP) assemblies of two Rhabdoviruses: rabies virus (RABV) and vesicular stomatitis virus (VSV). The authors provide brief descriptions of the N and M proteins, which form the RNP within the virions, and L and P, which are components of the viral transcription machinery. The authors also compare the structures and interactions of N and P proteins within crystal structures and docked within cryo-EM structures of viruses. Altogether, I find this manuscript to be a succinct description of our current state of knowledge on the structure of Rhabdovirus RNPs. It is well written and cited, and will be informative to both experts on Rhabdoviruses and Mononegaviruses, as well as the broader virology community. I recommend publication of this review with the following minor revisions:

  1. In figure 1, it is hard to tell if the two panels are equivalently scaled or just very close. I suggest scaling them equivalently and only showing one scale bar. Also, the black scale bar blends into the background; increasing the thickness and adding a white outline around the bar and text would help.

  1. For figure 2, scale is also difficult to gauge. Each panel should have a scale bar. Also, for panel A, are subpanels i and ii the same scale? To my eye, it appears than subpanel ii is smaller; if so, they should be at the same scale.

  1. For figure 4, the viruses are not labelled. I assume that left is RABV and right is VSV from the figure title, but this is not explicitly stated. The viruses should be labelled in the figure. Again, are each panels at the same scale?

Author Response

We would like to thank the reviewer for his effort and help to improve our manuscript. Below is a point to point response to the reviewers comments. Our answers are written in blue and flanked by +++.

In figure 1, it is hard to tell if the two panels are equivalently scaled or just very close. I suggest scaling them equivalently and only showing one scale bar. Also, the black scale bar blends into the background; increasing the thickness and adding a white outline around the bar and text would help.

+++Line 51: 'at the same scale' has been added to the picture legend. Scale bars have been modified according to the reviewer's suggestion. +++

For figure 2, scale is also difficult to gauge. Each panel should have a scale bar. Also, for panel A, are subpanels i and ii the same scale? To my eye, it appears than subpanel ii is smaller; if so, they should be at the same scale.

+++We did not add scale bars to this panel as the images are shown in perspective view. We added a statement in the figure legend to clarify that the images are not to scale: line 82 Panels i and ii are not at the same scale.+++

For figure 4, the viruses are not labelled. I assume that left is RABV and right is VSV from the figure title, but this is not explicitly stated. The viruses should be labelled in the figure. Again, are each panels at the same scale?

+++We added the labels to the figure as requested. Scale of A and B are comparable, in C, scale has been adjusted to allow for a better overview of the curvature. Exact scale is not given due to the display in perspective view. +++

Reviewer 2 Report

Riedel and colleagues have submitted a manuscript on the Rhabdoviral ribonucleoprotein. Details on the mechanisms of rhabdoviral transcription and replication have been uncovered over the last four decades. In the last two decades of this time period, structures for each viral protein have been determined with culminated in detail of not only the individual proteins, but also of protein complexes and the intact virion structures of VSV and RABV (the work of the authors, here). In this review, rhabdoviral proteins are discussed with respect to function, structure and placement in the virion, specifically with respect to the RNP. Below are a few points to be considered and/or corrected prior to publication.

Line 12: As noted in the introduction (line 29), not all rhabdoviruses have a bullet shape, with some having bacilli-form shape. Unless the intention in line 12 is toward isolating the structures of VSV and Rabies (as presented in the later part of the review), the abstract should concur with this point.

Line 33: One point that the authors might consider is the presentation of the concept of ribonucleoprotein. Historically, ribonucleoprotein is generically used to define a nucleic acid/protein complex. In the case of rhabdoviruses, the term “ribonucleoprotein complex” has been used to define the capsids (intact polymerized N-RNA) as well as the larger complexes as described here. The authors may consider to use the term ribonucleoprotein complex as an alternative to ribonucleoprotein, at their discretion.

Line 62: “Higher order N-protein assemblies are stabilized by an N- and a C-terminally protruding domain.” These areas are more like glorified extensions or loops, that lack tertiary structure in the absence of their binding partners. Extension or loop, respectively, better describes these stabilizing entities.

Figure 2: To aid the reader, the N- and a C-terminally protruding parts should be labeled.

Line 109+: Because the authors are describing a body of work and interchangeably describing proteins from RABV and VSV. It might be confusing to a general audience when mixing the two without clarification. For example: “These are the N-terminal domain (PNTD aa 1-52), the N-terminal intrinsically disordered domain (PNID aa 53-90), the central domain (PCED aa 91-130), the C-terminal intrinsically disordered domain (PCID aa 131-195) and the C-terminal domain (PCD aa 195-296).” These define the primary sequence distribution of RABV P. The next line states: “Atomic structures of PCED (pdb: 2fqm [38])... and PCD (pdb: 1vyi [39])”. Here, 2fqm references the VSV PCED structure which is structured to aa 172. 1vyi references RABV PCD. The simple remedy is to define in the sentence that amino acid numbering refers to RABV. In subsequent structure references, perhaps (pdb: 2fqm [38], VSV) would help.

A second point, the equivalent structure of PCED has been published for RABV (id :3L32; Structure of the dimerisation domain of the rabies virus phosphoprotein,  Ivanov, I., Crepin, T., Jamin, M., Ruigrok, R.W.H., JVI 2010 DOI: 10.1128/JVI.02557-09), listing both would be more comprehensive; Likewise for VSV PCD (2K47; Solution structure of the C-terminal N-RNA binding domain of the Vesicular Stomatitis Virus Phosphoprotein   Ribeiro, E.A. et al. DOI: 10.2210/pdb2K47/pdb).

Line 128: “5. L-protein structure and function” There is no real discussion of the structure here. The authors could describe the overall modular layout of the polymerase, with a brief discussion of the core and appendage domains/arrangement. Also, the structure of the RABV L protein has recently been published (Structure of a rabies virus polymerase complex from electron cryo-microscopy. Horwitz, J.A., Jenni, S., Harrison, S.C., Whelan, S.P.J.  (2020) Proc Natl Acad Sci U S A 117: 2099-2107. DOI:10.1073/pnas.1918809117).

Line 129: RE: compound, did the authors mean component?

Line 137: the authors should define what No is, as the general audience might not know that this is the monomeric N encapsidation precursor.

Line 149: “VSV RNPs containing the genomic strand are preferentially packaged” Did the authors mean to state that the (-) sense strand is packed into virions?

Line 151: “Highly biased replication provides for 98% of genomic RNAs.” Could the authors expand on this statement, it would help to define what they mean here. Is the intent to state that there is a higher replication copy number (-) versus (+) sense genomes, etc? Generically (-) and (+) sense copies are genomic in nature. A little expansion on the concepts should clean this up nicely.

Figure 4: It would be helpful to the reader to define RABV (left) and VSV (right) in the legend and/or labeling in the Figure, as was done in Figure 2. Likewise, label the individual panels and calling them out in the legend would be good for the reader.

Author Response

We would like to thank the reviewer for his/her helpful comments to improve our manuscript. Below is a point to point response to the issues raised. Our answers are provided in blue and framed by +++. 

Line 12: As noted in the introduction (line 29), not all rhabdoviruses have a bullet shape, with some having bacilli-form shape. Unless the intention in line 12 is toward isolating the structures of VSV and Rabies (as presented in the later part of the review), the abstract should concur with this point.

+++line 12: 'or bacteroid' has been added+++

Line 33: One point that the authors might consider is the presentation of the concept of ribonucleoprotein. Historically, ribonucleoprotein is generically used to define a nucleic acid/protein complex. In the case of rhabdoviruses, the term “ribonucleoprotein complex” has been used to define the capsids (intact polymerized N-RNA) as well as the larger complexes as described here. The authors may consider to use the term ribonucleoprotein complex as an alternative to ribonucleoprotein, at their discretion.

+++line 36: ribonucleoprotein has been replaced by ribonucleoprotein complex (RNP), and also the title has been adapted.+++

Line 62: “Higher order N-protein assemblies are stabilized by an N- and a C-terminally protruding domain.” These areas are more like glorified extensions or loops, that lack tertiary structure in the absence of their binding partners. Extension or loop, respectively, better describes these stabilizing entities.

+++line 70: domain has been replaced by extension+++

Figure 2: To aid the reader, the N- and a C-terminally protruding parts should be labeled.

+++These parts have been labeled in figure 2 A ii.+++

Line 109+: Because the authors are describing a body of work and interchangeably describing proteins from RABV and VSV. It might be confusing to a general audience when mixing the two without clarification. For example: “These are the N-terminal domain (PNTD aa 1-52), the N-terminal intrinsically disordered domain (PNID aa 53-90), the central domain (PCED aa 91-130), the C-terminal intrinsically disordered domain (PCID aa 131-195) and the C-terminal domain (PCD aa 195-296).” These define the primary sequence distribution of RABV P. The next line states: “Atomic structures of PCED (pdb: 2fqm [38])... and PCD (pdb: 1vyi [39])”. Here, 2fqm references the VSV PCED structure which is structured to aa 172. 1vyi references RABV PCD. The simple remedy is to define in the sentence that amino acid numbering refers to RABV. In subsequent structure references, perhaps (pdb: 2fqm [38], VSV) would help.

+++We adapted the nomenclature according to the reviewer's suggestion in lines 122-129.+++

A second point, the equivalent structure of PCED has been published for RABV (id :3L32; Structure of the dimerisation domain of the rabies virus phosphoprotein, Ivanov, I., Crepin, T., Jamin, M., Ruigrok, R.W.H., JVI 2010 DOI: 10.1128/JVI.02557-09), listing both would be more comprehensive; Likewise for VSV PCD (2K47; Solution structure of the C-terminal N-RNA binding domain of the Vesicular Stomatitis Virus Phosphoprotein Ribeiro, E.A. et al. DOI: 10.2210/pdb2K47/pdb).

+++We thank the reviewer for his help and have included the appropriate references in line 126 and 127+++

Line 128: “5. L-protein structure and function” There is no real discussion of the structure here. The authors could describe the overall modular layout of the polymerase, with a brief discussion of the core and appendage domains/arrangement. Also, the structure of the RABV L protein has recently been published (Structure of a rabies virus polymerase complex from electron cryo-microscopy. Horwitz, J.A., Jenni, S., Harrison, S.C., Whelan, S.P.J. (2020) Proc Natl Acad Sci U S A 117: 2099-2107. DOI:10.1073/pnas.1918809117).

+++We have modified the paragraph describing the L-protein as follows starting from line 149:
It can be divided into three catalytic and two structural domains. The RdRp and the capping domain form the core of the structure. In the absence of P, the remaining three domains – connector domain, methyltransferase domain and C-terminal domain – have no fixed position with regard to the core of the structure [6,40,41,80]. P-protein segments binding to L-protein that can be structurally resolved are aa 49-56 (interaction with C-terminal domain), 82-89 (binding between C-terminal domain and RdRp) and 94-105 (binding between connector and C-terminal domain) for VSV [40]. For RABV, a 37 aa segment of P-protein, ending with Glu87, could be identified bound to the connector, the RdRp and the C-terminal domain of L-protein [41]. +++

Line 129: RE: compound, did the authors mean component?

+++We thank the reviewer for pointing out this mistake and have replaced compound by component. +++

Line 137: the authors should define what No is, as the general audience might not know that this is the monomeric N encapsidation precursor.

+++We have added a definition of N0 in lines 134 and 159.+++

Line 149: “VSV RNPs containing the genomic strand are preferentially packaged” Did the authors mean to state that the (-) sense strand is packed into virions?

+++
We added 'genomic, negative sense strand' in line 170 for clarification.+++

Line 151: “Highly biased replication provides for 98% of genomic RNAs.” Could the authors expand on this statement, it would help to define what they mean here. Is the intent to state that there is a higher replication copy number (-) versus (+) sense genomes, etc? Generically (-) and (+) sense copies are genomic in nature. A little expansion on the concepts should clean this up nicely.

+++This sentence has been modified as follows (line 211): 'The presence of 98% of negative sense genomes in RABV particles is a result of highly biased replication, favouring the generation of negative sense, genomic RNAs.'+++

Figure 4: It would be helpful to the reader to define RABV (left) and VSV (right) in the legend and/or labeling in the Figure, as was done in Figure 2. Likewise, label the individual panels and calling them out in the legend would be good for the reader.

+++The figure has been adapted as requested by the reviewer. In line 255, the figure legend has been adapted by adding the following sentence: (A) shows a view from the side, (B) a view from the membrane and (C) a view towards the RNA binding groove. +++

Reviewer 3 Report

In this review, the authors aim to review the components of the rhabdovirus ribonucleoprotein and its higher order structural assembly. In particular, the review provides an interesting comparison between rabies virus and VSV ribonucleoprotein complex based on recent cryo-EM reconstructions. My main concern about this review is that it refers mainly to literature dated before 2010 and failed to cite more recent and relevant research articles. Beside two references dated 2019 (ref. 35 and 79), including one by the authors (79), there are only 3 articles and 1 review dated of 2015 and very few articles dated between 2010 and 2015 (see below). My recommendation is that the review should be improved in that sense before publication.

Major points

  • pg 5 - Residue numbers that are given here are for rabies virus. Up to this point, the authors discuss the properties of both rabies virus and VSV, which gives the feeling that the numbers are the same for both proteins. They should state that numbers are given as example for indicating the relative size of the different regions or they should remove them.
  • pg 5 - The authors state that P-protein is made of 5 structural modules of which two are disordered. If one considers the isolated P protein, I would rather consider that there are only 4 modules. The entire N-terminal region (1-90), which they separate in one structured and one disordered modules, is entirely disordered in isolated P, with only small amount of transient secondary structures in the N-terminal N0-binding region as shown by Leyrat et al. 2011a (doi: 10.1002/pro.587) and Leyrat et al. 2012 (doi: 10.1016/j.jmb.2012.07.003) for VSV and more recently by Jespersen et al. 2019 (doi: 10.1016/j.jmb.2019.10.011) for rabies virus. When the protein interacts with its partners, some of the disordered regions fold as shown for the VSV N0-binding region by Leyrat et al. 2011b (doi: 10.1371/journal.ppat.
    1002248), for the VSV and rabies L binding region by Jenni et al. 2020 (doi: 10.1016/j.celrep.2019.12.024), by Horwitz et al. 2020 (doi: 10.1073/pnas.1918809117) and by Gould et al. 2020 (doi: 10.1128/JVI.01729-19) or for the rabies LC2 binding region by Jespersen et al. 2019.
  • pg 5 - line 120 - The authors state that a dimer of P binds one molecule of unassembled N0 via residues in PNTD. In an initial biophysical characterization of the N0-P complex, Peluso and Moyer in 1988 (DOI: 10.1016/0042-6822(88)90477-1) found 1:1 stoichiometry and more recently, Yabukarski et al. 2016 (doi: 10.1016/j.jmb.2016.04.010) demonstrated that stoichiometry of 2P:1N or 2P:2N could be obtained depending on the experimental concentrations of the proteins.
  • pg 5 - line 120 - There is a structure of the complex between VSV N0 and part of PNTD by Leyrat et al. 2011b (doi: 10.1371/journal.ppat.1002248), which could be mentioned
  • pg 5 - line 123 - the authors cite reference 52 and 53 to support that PNTD and PNID interact with L. There is no mention of an interaction between PNTD and L in reference 52 by Ivanov et al. 2010 (structure of PCED of rabies virus). Castel et al. demonstrated interaction between rabies PNID and L in 2009 (doi: 10.1128/JVI.00977-09).
  • pg 5 - line 123 - There are two recent structure of complexes between L and P of rabies virus by Horwitz et al. 2020 (doi: 10.1073/pnas.1918809117) and of VSV by Jenni et al. 2020 (doi: 10.1016/j.celrep.2019.12.024) that show part of PNID region bound to L
  • pg 5 - line 125 - again reference 52 is incorrect and does not shown that PCD binds to the N-RNA complex
  • pg 5 - line 126-127 - the authors indicate that P antagonizes the interferon system and cite several targeted proteins. These interactions again are for rabies virus P but not for VSV. This should be stated, as far no interaction with components of the interferon pathway has been found with VSV P. For rabies virus, interactions with other host partners have also been found and could be mentioned, including ribosomal protein L9 by Li et al. 2016, with the mitochondrial complex I by Kammouni et al in 2015, with focal adhesion kinase by Fouquet et al. in 2015 or among others with LC2 dynein binding chain by Raux et al. in 2000 and Jespersen et al. in 2019.
  • pg 5 - line 128-135 - The authors cite the initial structure of VSV L reported by Liang et al. in 2014 but do not mention et al. the two recent publications of the L/P of rabies and VSV by Horwitz et al. 2020 (doi: 10.1073/pnas.1918809117) and of VSV by Jenni et al. 2020 (doi: 10.1016/j.celrep.2019.12.024). There are also X-ray structures of VSV L domains by Gould et al. in 2020 and by Paesen in 2015 (doi: 10.1038/ncomms9749).
  • pg 5 - line 138 - The authors suggest that charges play a central role in the interaction between PNTD and the RNA binding groove of N0, which implies that PNTD binds in the RNA binding groove. In support of this they provides two articles from 2001 and 1986 and a review from 2011, but they do not mention the X-ray structure of the N0-PNTD complex of VSV (Leyrat et al. 2011b) which clearly shows that the interaction is driven by hydrophobic residues and that PNTD does not occupy the position of the RNA on the N0 protein. Similar types of interaction between PNTD and N0 have been shown also for different viruses in the Paramyxoviridae, Pneumoviridae and Filoviridae family. The structure of this complex (N0-PNTD) is conserved between the different families. Concerning the difference in affinity between P and RNA for N0, it was experimentally demonstrated for Ebola virus by Kirchdoerfer in 2015 (/dx.doi.org/10.1016/j.celrep.2015.06.003) and very recently the assembly of nucleocapsid of measles virus was reconstituted in vitro by Milles et al. in 2016 (doi: 10.1002/anie.201602619.), clearly showing that it is driven by a difference in affinity.

Minor points

  • pg 1 - should also give the estimated number of G molecules in the virion to be complete
  • pg 2 - the symbols showing M and N proteins in two blue shades are difficult to visualize on the picture. Should use different colors to make that clearer
  • pg 3 - In figure 2B, it is difficult to see the residues highlighted in two different shades of green. Should show a zoom of the RNA binding cleft
  • pg 4 - line 96 - change "motive" into "motif"

Author Response

We thank the reviewer for his / her detailed report and highly appreciated help in improving our manuscript, especially regarding the structure and function of P protein. Below is a point to point response to all the issues raised. Our answers / modifications to the manuscript are indicated by +++ and in blue. 

Major points

pg 5 - Residue numbers that are given here are for rabies virus. Up to this point, the authors discuss the properties of both rabies virus and VSV, which gives the feeling that the numbers are the same for both proteins. They should state that numbers are given as example for indicating the relative size of the different regions or they should remove them.

+++line 122ff: We specified the species for the residues given. +++

pg 5 - The authors state that P-protein is made of 5 structural modules of which two are disordered. If one considers the isolated P protein, I would rather consider that there are only 4 modules. The entire N-terminal region (1-90), which they separate in one structured and one disordered modules, is entirely disordered in isolated P, with only small amount of transient secondary structures in the N-terminal N0-binding region as shown by Leyrat et al. 2011a (doi: 10.1002/pro.587) and Leyrat et al. 2012 (doi: 10.1016/j.jmb.2012.07.003) for VSV and more recently by Jespersen et al. 2019 (doi: 10.1016/j.jmb.2019.10.011) for rabies virus. When the protein interacts with its partners, some of the disordered regions fold as shown for the VSV N0-binding region by Leyrat et al. 2011b (doi: 10.1371/journal.ppat.
1002248), for the VSV and rabies L binding region by Jenni et al. 2020 (doi: 10.1016/j.celrep.2019.12.024), by Horwitz et al. 2020 (doi: 10.1073/pnas.1918809117) and by Gould et al. 2020 (doi: 10.1128/JVI.01729-19) or for the rabies LC2 binding region by Jespersen et al. 2019.

+++line 122-126 have been adapted as follows:
P-protein (34kD) is a central (co-) factor in the replication cycle of Rhabdoviruses. It can be divided into 4 domains. These are the N-terminal domain (PNTD aa 1-90, RABV), which is disordered but adopts defined conformations upon interaction with N or L [37–42], the central domain (PCED aa 91-130, RABV), the C-terminal intrinsically disordered domain (PCID aa 131-195, RABV) and the C-terminal domain (PCD aa 195-296, RABV). +++

pg 5 - line 120 - The authors state that a dimer of P binds one molecule of unassembled N0 via residues in PNTD. In an initial biophysical characterization of the N0-P complex, Peluso and Moyer in 1988 (DOI: 10.1016/0042-6822(88)90477-1) found 1:1 stoichiometry and more recently, Yabukarski et al. 2016 (doi: 10.1016/j.jmb.2016.04.010) demonstrated that stoichiometry of 2P:1N or 2P:2N could be obtained depending on the experimental concentrations of the proteins.

+++line 133-135: We have changed the sentence as follows to correct our mistake:
P-protein is preventing the self-assembly and RNA-binding of N-protein (monomeric N-protein not bound to RNA = N0), with one N-protein bound to one or two P-proteins via residues in PNTD [54–59],...+++

pg 5 - line 120 - There is a structure of the complex between VSV N0 and part of PNTD by Leyrat et al. 2011b (doi: 10.1371/journal.ppat.1002248), which could be mentioned

+++line 128-129: We have included this structure: ...–, as well as parts of VSV PNTD with N [37],... +++

pg 5 - line 123 - the authors cite reference 52 and 53 to support that PNTD and PNID interact with L. There is no mention of an interaction between PNTD and L in reference 52 by Ivanov et al. 2010 (structure of PCED of rabies virus). Castel et al. demonstrated interaction between rabies PNID and L in 2009 (doi: 10.1128/JVI.00977-09).

+++line 220-221: We thank this reviewer for pointing out our mistake and have corrected it. +++

pg 5 - line 123 - There are two recent structure of complexes between L and P of rabies virus by Horwitz et al. 2020 (doi: 10.1073/pnas.1918809117) and of VSV by Jenni et al. 2020 (doi: 10.1016/j.celrep.2019.12.024) that show part of PNID region bound to L

+++line 220-221: These references have been added.+++

pg 5 - line 125 - again reference 52 is incorrect and does not shown that PCD binds to the N-RNA complex

+++The reference has been removed.+++

pg 5 - line 126-127 - the authors indicate that P antagonizes the interferon system and cite several targeted proteins. These interactions again are for rabies virus P but not for VSV. This should be stated, as far no interaction with components of the interferon pathway has been found with VSV P. For rabies virus, interactions with other host partners have also been found and could be mentioned, including ribosomal protein L9 by Li et al. 2016, with the mitochondrial complex I by Kammouni et al in 2015, with focal adhesion kinase by Fouquet et al. in 2015 or among others with LC2 dynein binding chain by Raux et al. in 2000 and Jespersen et al. in 2019.

+++We thank this reviewer for pointing out this lack of detail. The sentence has been adapted as follows:
line 223-226: The P-protein of RABV is interfering with the phosphorylation of IRF3 [65], retains activated STAT1 and STAT2 in the cytoplasm [50,66] and interacts with TRIM19 [67–69]. Interactions of RABV P-protein with additional cellular proteins have been described [39,70–73], but are beyond the scope of this review. +++

pg 5 - line 128-135 - The authors cite the initial structure of VSV L reported by Liang et al. in 2014 but do not mention et al. the two recent publications of the L/P of rabies and VSV by Horwitz et al. 2020 (doi: 10.1073/pnas.1918809117) and of VSV by Jenni et al. 2020 (doi: 10.1016/j.celrep.2019.12.024). There are also X-ray structures of VSV L domains by Gould et al. in 2020 and by Paesen in 2015 (doi: 10.1038/ncomms9749).

+++line 228-239: both publications have been included+++

pg 5 - line 138 - The authors suggest that charges play a central role in the interaction between PNTD and the RNA binding groove of N0, which implies that PNTD binds in the RNA binding groove. In support of this they provides two articles from 2001 and 1986 and a review from 2011, but they do not mention the X-ray structure of the N0-PNTD complex of VSV (Leyrat et al. 2011b) which clearly shows that the interaction is driven by hydrophobic residues and that PNTD does not occupy the position of the RNA on the N0 protein. Similar types of interaction between PNTD and N0 have been shown also for different viruses in the Paramyxoviridae, Pneumoviridae and Filoviridae family. The structure of this complex (N0-PNTD) is conserved between the different families. Concerning the difference in affinity between P and RNA for N0, it was experimentally demonstrated for Ebola virus by Kirchdoerfer in 2015 (/dx.doi.org/10.1016/j.celrep.2015.06.003) and very recently the assembly of nucleocapsid of measles virus was reconstituted in vitro by Milles et al. in 2016 (doi: 10.1002/anie.201602619.), clearly showing that it is driven by a difference in affinity.

+++We thank this reviewer for clarifying this point and have adapted the paragraph as follows: line 242-244: It has been shown that PNTD interacts with N0 via hydrophobic residues, and that the interaction interface is not located in the RNA binding groove [37].+++

Minor points

pg 1 - should also give the estimated number of G molecules in the virion to be complete

+++This number has been added as requested in line 44.+++

pg 2 - the symbols showing M and N proteins in two blue shades are difficult to visualize on the picture. Should use different colors to make that clearer

+++We have adjusted the color for better visualisation.+++

pg 3 - In figure 2B, it is difficult to see the residues highlighted in two different shades of green. Should show a zoom of the RNA binding cleft

+++We have adapted the figure to show a zoom into the RNA binding cleft.+++

pg 4 - line 96 - change "motive" into "motif"

+++We have corrected this mistake. +++